# A New Composite Structure of PEDOT/PSS: Macro-Separated Layers by a Polyelectrolyte Brush

**DOI:** 10.3390/polym12020456

**Published:** 2020-02-16

**Authors:** Keita Yasumoro, Yushi Fujita, Hideki Arimatsu, Takuya Fujima

**Affiliations:** 1Department of Mechanical Engineering, Tokyo City University, Tokyo 158-8557, Japan; g1881062@tcu.ac.jp (K.Y.); g1991001@tcu.ac.jp (H.A.); 2Advanced Research Laboratories, Tokyo City University, Tokyo 158-0082, Japan; g1981041@tcu.ac.jp

**Keywords:** PEDOT, transparent, conductive polymer, polyelectrolyte brush

## Abstract

Polyethylene dioxythiophene and polyethylene sulfonic acid (PEDOT/PSS) composite is gathering attention as an organic transparent conductive film material. However, it requires a core-shell structure in which conductive PEDOT is covered with insulating PSS. Providing film formability and a carrier to PEDOT, the PSS shell hinders carrier conduction as an insulating barrier. In this study, we realized that creating a macro-separated PEDOT/PSS composite by using a polyelectrolyte brush substrate and in-situ PEDOT polymerization without the PSS barrier increases durability and conductivity in comparison with commercially available PEDOT/PSS film, achieving a conductivity of 5000–6000 S/cm.

## 1. Introduction

The increasing demand for transparent conductive film is uncontroversial as devices like liquid crystal displays and solar panels are widely popularized. Indium-tin-oxide (ITO) is mainly used as a conductor due to its high electrical conductivity and transparency [1]. However, a vacuum process for coating on a base material, vapor deposition, increases the ITO manufacturing cost. Additionally, a concern arises as regards a component of ITO, indium, not only about cost but also stable supply because it is a rare metal.

Among the conductive polymers [2,3,4], a polyethylene dioxythiophene and polyethylene sulfonic acid (PEDOT/PSS) composite is widely researched because of its high electrical conductivity and chemical stability [5,6]. PSS provides a conductive carrier and film formability to PEDOT, which only has a pi-conjugated conductive structure without any carrier; it is also difficult to form a film on a substrate without any stabilizer. PSS, therefore, is added to provide a conductive carrier as a dopant and film formability as a stabilizer [7].

In most cases, the PEDOT/PSS composite is provided as an aqueous dispersion to apply directly to a substrate, where PEDOT reportedly forms a core-shell structure with a PSS shell [8]. The PSS, an insulating material, acts as a barrier between PEDOT chains, decreasing the conductivity [9]. Many studies to reduce the PSS and improve the molecular packing within formed films have been carried out so far by adding various chemical compounds such as ethylene glycol [10,11,12]. On the other hand, studies to approach the intrinsic structure of PEDOT/PSS to resolve the insulating barrier are quite rare, although there are a few without PSS [13,14] like the authors’ previous work [15] using a hierarchical nonporous layer glass [16,17].

In this work, we tried to change the structure of the PEDOT/PSS composite from a micro-separated core-shell one to a macro-separated layered one to avoid the barrier. That is, we designed a PEDOT film on a PSS layer covalently bonded on to a substrate. The PSS forms a polyelectrolyte brush (PEB) [18] where the polyelectrolyte chain stretches away from a substrate in water due to the osmotic pressure caused by the dissociated ions.

## 2. Materials and Methods

The method reported by Tran and Auroy [19,20] was used to prepare the polyelectrolyte brush. Chemicals (toluene, acetic anhydride, sulfuric acid, 1,2-dichloroethane) were purchased from Fujifilm Wako Pure Chemical Industries and used without further purification. We used a soda-lime glass slide (Matsunami Glass Co. Ltd., Osaka, Japan) as a substrate after ultrasonic cleaning in acetone and purified water before use. First, we performed vacuum ultraviolet light-ozone (VUV-ozone) treatment on the substrate and immersed it in a toluene solution of anion-polymerized SiCl_3_-terminated polystyrene (molecular weight = 47,000) for 1 h. This sample was heated in a vacuum at 160 °C for one day to accelerate the dehydration–degeneration reaction, thereby obtaining a polystyrene (PS) brush. We heated the prepared PS brush at 60 °C for 1 h in a mixture of 3.4 mL of acetic anhydride and 1.2 mL of sulfuric acid diluted with 16.4 mL of 1,2-dichloroethane to add a sulfo group to the polystyrene chain to convert the sample into a polystyrene sulfuric acid (HPSS) brush. Finally, to chemically stabilize the PSS brush, the sample was immersed in a 0.5 mol dm^−3^ aqueous solution of sodium hydrogen carbonate for one day to replace the counterion from H^+^ to Na^+^.

We referred to the method reported by Mulfort et al. [21] for depositing a PEDOT transparent conductive film on the PEB by in situ polymerization. Iron (III) chloride (>95%) and 3,4 Ethylenedioxythiophene (EDOT, >98%) were purchased from Fujifilm Wako Pure Chemical Industries and Tokyo Chemical Industry, respectively, and used without further purification. We added 0.26 mass % of EDOT to water and stirred for 15 h, then added NaPSS brush substrate and 0.99 mass % of iron (III) chloride and stirred for 48 h. After that, it was washed with purified water to obtain a PEDOT/PEB film.

We also prepared a commercial PEDOT/PSS transparent conductive film for comparison with the obtained PEDOT/PEB sample. A 1.0 wt % PEDOT/PSS aqueous dispersion of high-conductivity grade (Aldrich 768642) was purchased from Sigma-Aldrich and spin-coated at 500 rpm on untreated glass to use as the comparative sample.

The samples were subjected to ultrasonication (37 kHz, 10 W) in water for 10 min to verify their durability, and the changes in conductivity and Raman scattering before and after the ultrasonication were examined.

The conductivity of the obtained PEDOT/PEB sample was measured by a four-terminal method using an LCR meter (ZM2376, NF Corporation, Yokohama, Japan) under the conditions of an applied voltage of 5 V and a measurement frequency of 50 mHz to 1 kHz. The Raman scattering was performed using a triple Raman spectrometer, (T64000, HORIBA, Kyoto, Japan).

We also characterized the samples by the water contact angle using a contact angle meter (PCA-1, Kyowa Interface Science Co., Ltd., Saitama, Japan) Therein, the water droplet volume was set to 3 µL to suppress the effects of gravity and water droplet evaporation, which also complies with JIS.

## 3. Results and Discussion

The obtained PEDOT/PEB sample had a homogeneous color tone throughout, and a pale bluish color peculiar to PEDOT appeared. This is consistent with the visible light transmittance spectrum shown in Figure 1. Although the spectrum is largely similar to that of the untreated glass, the transmittance on the long-wavelength side is slightly lowered and matches the bluish color tone described above. It is considered that the transmittance decrease on the long-wavelength side is caused by absorption by polaron, a conductive carrier in PEDOT [22,23]. On the other hand, the sample coated with the commercially available PEDOT/PSS dispersion had a significantly lower transmittance and a darker tone than the PEDOT/PEB sample.

We also determined the thickness of the PEDOT/PEB sample as 15–20 nm by using a secondary electron image (SEI, Figure 2). The SEI indicated that the PEDOT/PEB film is in firm contact with the glass substrate.

Figure 3 shows photographs of the PEDOT/PEB sample and a commercially available PEDOT/PSS coated sample, which were exposed to ultrasonic waves in water. While the sample to which the aqueous dispersion was applied easily peeled off from the substrate, PEDOT/PEB did not peel off, and there was no change in appearance.

Figure 4 shows the conductivity spectra of the samples before and after ultrasonic treatment. As for the sample coated with commercially available PEDOT/PSS, resistivity increased significantly because the PEDOT was peeled off as described above. On the other hand, the PEDOT/PEB sample exhibited sheet resistance equivalent to that of the commercially available PEDOT/PSS film before ultrasonication and maintained the electrical property and appearance even after the ultrasonic treatment.

Figure 1 shows that the PEDOT/PEB has an absorbance of about 1/10 that of a commercial PEDOT/PSS film, according to Lambert–Beer law. Since the sheet resistivity is equivalent as shown in Figure 4, PEDOT/PEB realized about ten times the volume conductivity of the other. The volume conductivity of the PEDOT/PEB is also estimated to be very high, about 5000–6000 S/m, from Figure 2 and Figure 4. The estimated value is also an order of magnitude higher than the specification of the commercial PEDOT/PSS (ca. 170 S/cm).

We performed Raman scattering measurements to investigate the change in PEDOT molecules’ abundance. The scattering spectrum showed a series of three peaks appearing at 1400–1600 cm^−1^ derived from PEDOT molecules [24,25], as shown in Figure 5. The decrease in these peak intensities indicated that a small part of the excess PEDOT molecules was removed by ultrasonic treatment.

The water contact angles on PEDOT/PEB and PEDOT/PSS samples were 67° and less than 10° (too small to define), respectively, indicating significantly different surface properties. The latter contact angle comes from the water-soluble PSS exposed to the sample surface because the coated PEDOT/PSS colloid has a PSS shell. On the other hand, the former is due to the PEDOT molecules, which are hardly soluble in water [26], covering the sample surface, indicating that a macroscopically separated structure in which the PEDOT layer overlaps the PSS brush layer has been formed.

The result of the ultrasonic treatment damaging neither the film nor the conductive property suggests that the PEDOT chain is firmly bonded to the PEB substrate in the PEDOT/PEB sample. That is, a strong bundle between PEDOT and PSS chains, which reportedly exists in the PEDOT/PSS core-shell structure [27,28], was formed on the PEB.

Consequently, the PEDOT/PEB sample is thought to have achieved a macro-separated configuration of the composite of PEDOT and PSS; the in situ polymerized PEDOT chains bundled with the PSS chains covalently bonded to the glass slide, and the PEDOT chain was deposited to cover the PEB as shown in Figure 6. This is consistent with the results of the ultrasonic treatment and water contact angle.

In this case, the PEDOT is stably bonded to the substrate and receives a charge from the PSS. Furthermore, in the PEDOT-rich layer deposited on the PEB, carrier mobility should be improved due to the absence of the insulation barrier of a PSS shell in the core-shell structure. This resulted in a thinner film (higher optical transmittance) with higher electrical conductivity than commercially available PEDOT/PSS film.

## 4. Conclusions

In this study, we tried to achieve a macro-separated structure of a PEDOT/PSS composite film other than the micro-separated structure of the conventional core-shell type PEDOT/PSS colloid coating. Therefore, we grafted PSS molecules onto a glass substrate to form a polyelectrolyte brush (PEB) on which we deposited PEDOT by in situ polymerization. The resulting PEDOT/PEB sample exhibited high water resistance and achieved a higher conductivity, 5000–6000 S/cm, than a film of a commercially available PEDOT/PSS colloidal dispersion. The characteristics are thought to come from the strong bundle of PEDOT chains with PSS covalently bonded to the substrate and the PEDOT-rich layer without the insulating barrier of the PSS shell.

## Figures and Tables

**Figure 1 polymers-12-00456-f001:**
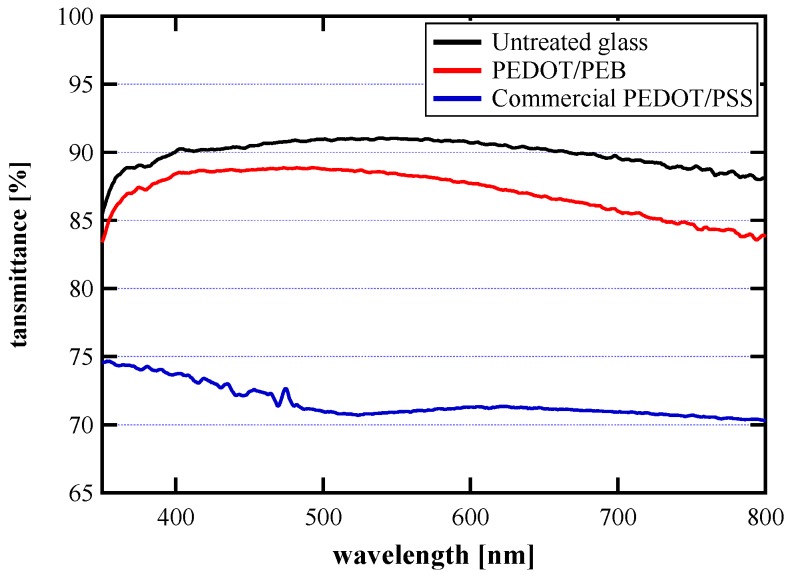
Optical transmittance spectra of the polyethylene dioxythiophene on polyelectrolyte brush (PEDOT/PEB) sample, commercial polyethylene dioxythiophene and polyethylene sulfonic acid (PEDOT/PSS) film and untreated glass. The PEDOT/PEB sample exhibited remarkably high transparency.

**Figure 2 polymers-12-00456-f002:**
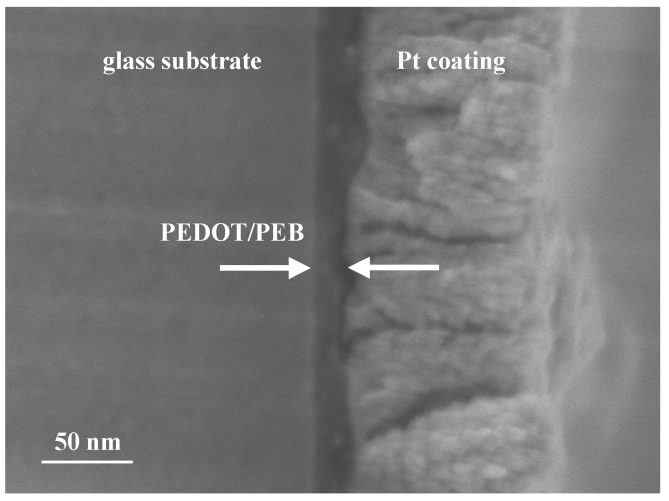
SEM micrograph of a cross-section of the PEDOT/PEB film. A porous layer on the right side is Pt coating for SEM observation.

**Figure 3 polymers-12-00456-f003:**
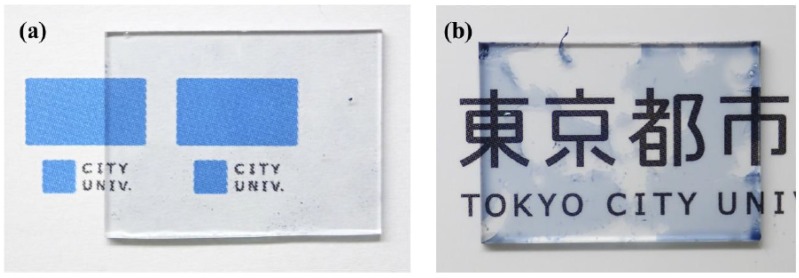
Appearance of (**a**) the PEDOT/PEB sample and (**b**) commercial PEDOT/PSS film after ultrasonic treatment in water. The conventional PEDOT/PSS film peeled off heterogeneously other than the PEDOT/PEB sample.

**Figure 4 polymers-12-00456-f004:**
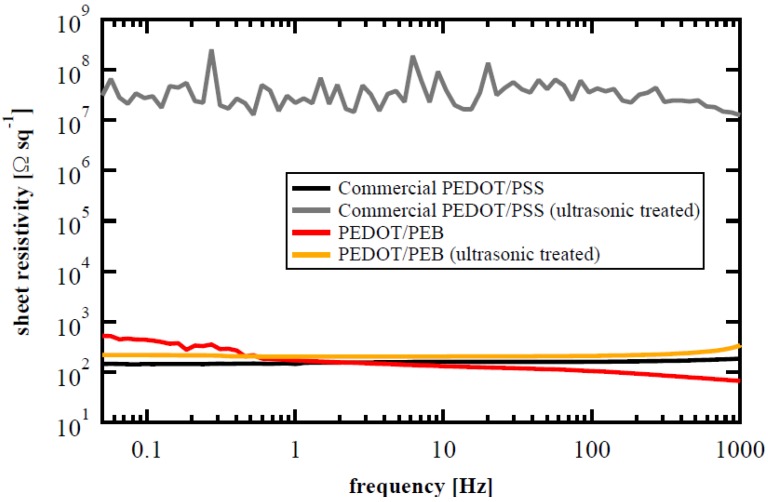
Sheet resistivity spectra of the PEDOT/PEB sample and commercial PEDOT/PSS film before and after ultrasonic treatment in water. The PEDOT/PEB sample maintained its conductivity even after ultrasonic treatment.

**Figure 5 polymers-12-00456-f005:**
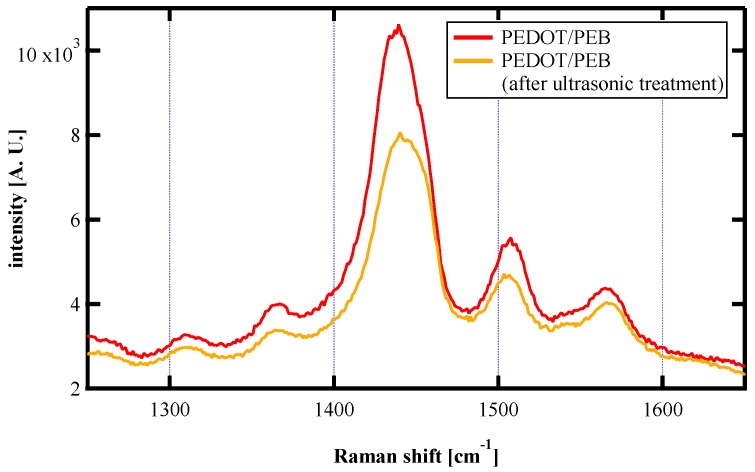
Raman spectra of the PEDOT/PEB sample before and after ultrasonic treatment.

**Figure 6 polymers-12-00456-f006:**
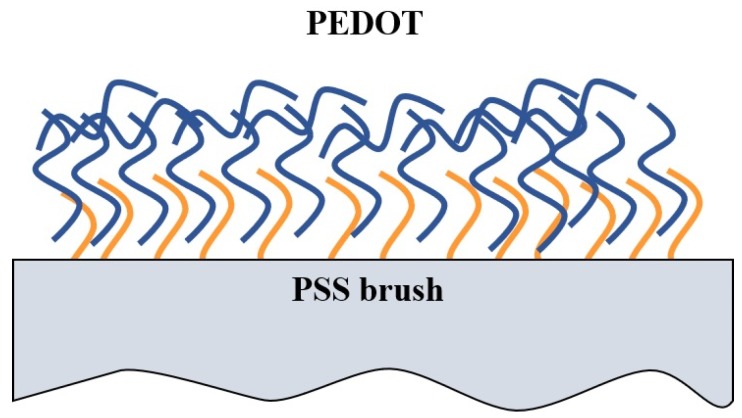
Schematic diagram of the PEDOT/PEB composite.

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
