# Peer review of "A New Composite Structure of PEDOT/PSS: Macro-Separated Layers by a Polyelectrolyte Brush"

_polymers, 2020, doi:10.3390/polym12020456_

Round 1

Reviewer 1 Report

This manuscript describes an interesting approach for the in-situ synthesis of PEDOT over a previously coated PSS substrate, improving the stability in water compared to commercial PEDOT films. This approach could be very useful in technology where transparent and stable PEDOT films are needed, thus I recommend this paper for publication in the present form.

Author Response

The authors appreciate the referee's evaluation of the manuscript, and also hope that this work will be useful in the field of the transparent conductive film.

Reviewer 2 Report

In this work, the authors realized that creating a PEDOT: PSS composite by using a polyelectrolyte brush substrate and in-situ PEDOT polymerization. In my opinion, this work is important which may be publishable on Polymers. Some concerns should be addressed before publishing.

Regarding commercial PEDOT: PSS, what’s the whole name? Like PH500, PH1000 or 4083? What’s the thickness of different PEDOT: PSS layers? Since transmittance is related with film thickness. Film morphology is critical, AFM images of different PEDOT: PSS layers can increase the quality of the manuscript. In the introduction part, some refs (Org. Electron. 2017, 41, 179-185; Sci. Rep., 2017, 7, 16468) related on transparent electrodes are recommended.

Author Response

First, the authors appreciate the referee's evaluation that this work is important. The authors then respond to the concerns raised by the referees as follows:

1) Regarding commercial PEDOT: PSS, what's the whole name?

   The authors left out the product name of the commercial PEDOT in the previous manuscript and thank the referee for pointing it out. The product name has been added to "Materials and Methods" in this revision.

2) What's the thickness of different PEDOT: PSS layers? Since transmittance is related with film thickness.

   The authors observed a cross-section of the PEDOT/PEB sample with SEM. The micrograph has been added as Figure 2 to clarify the film thickness. Also, as pointed out by the referee, there is a direct relationship between film thickness and transmittance. More specific estimates for PEDOT/PEB sample and commercial PEDOT/PSS film thickness have been added to "Results & Discussion." The estimated volume conductivity of the PEDOT/PEB has also been added to "abstract" and "Conclusions."

3) Film morphology is critical, AFM images of different PEDOT: PSS layers can increase the quality of the manuscript.

   The authors agree with the referee that information on membrane morphology is important and will make this manuscript better. Unfortunately, however, the authors have not conducted AFM measurements on this PEDOT system and will likely take some time to make new investigations and obtain reliable data. This work focuses on material creation to realize a macro-separated PEDOT/PSS composite with high conductivity, and the authors are planning that details of membrane morphology and conduction mechanism will be reported in the next paper.

4) In the introduction part, some refs (Org. Electron. 2017, 41, 179-185; Sci. Rep., 2017, 7, 16468) related on transparent electrodes are recommended.

   The authors appreciate the referee's recommendation. However, in the introduction of this manuscript, the first paragraph mentions the ITO, and the second paragraph does conductive polymers to focus on the main topic of this manuscript, PEDOT. Inserting the topic of silver nanowires recommended by the referee into that would result in an unnatural configuration. Therefore, in the next paper, the authors will set up a paragraph that reviews a wide range of transparent conductive films including silver nanowires, and would like to refer the recommended papers there.